# Comparison and Harmonization of Different Semi-Automated and Automated qRT-PCR Assays in the Assessment of SARS-CoV-2

**DOI:** 10.3390/v14102239

**Published:** 2022-10-12

**Authors:** Sascha Dierks, Karin Thiele, Wolfgang Bohne, Raimond Lugert, Michael Weig, Uwe Groß, Nicolas von Ahsen, Julie Schanz, Andreas Fischer, Moritz Schnelle

**Affiliations:** 1Department of Clinical Chemistry, University Medical Center Göttingen, 37075 Göttingen, Germany; 2Interdisciplinary UMG Laboratory, University Medical Center Göttingen, 37075 Göttingen, Germany; 3Medical Microbiology and Virology, University Medical Center Göttingen, 37075 Göttingen, Germany; 4Division Vascular Signaling and Cancer, German Cancer Research Center (DKFZ), 69120 Heidelberg, Germany

**Keywords:** COVID-19, SARS-CoV-2, quantitative reverse transcriptase-polymerase chain reaction, cycle threshold value, viral load

## Abstract

In SARS-CoV-2 diagnostics, cycle threshold (Ct) values from qRT-PCRs semi-quantitatively estimate a patient’s viral load. However, relevant analytical differences between qRT-PCR assays are often neglected. This study was designed (i) to identify such differences between five commonly used assays and (ii) to demonstrate a straightforward strategy to harmonize them. QRT-PCRs for SARS-CoV-2 were carried out in 85 oropharyngeal swab samples using three fully automated (Alinity m, cobas^®^6800 and GeneXpert) and two semi-automated (genesig^®^ and RIDA^®^GENE) assays. Qualitative results (positive/negative) showed excellent comparability between the fully automated assays, but not between the Alinity m and semi-automated methods. Ct values significantly varied between all the methods, with the median values ranging from 22.76 (Alinity m) to 30.89 (RIDA^®^GENE) and 31.50 (genesig^®^), indicating the lowest sensitivity for semi-automated methods. Passing–Bablok analysis further revealed systemic biases. Assay-specific viral load concentration calculations—based on generated individual standard curves—resulted in much better comparability between the assays. Applying these calculations, significant differences were no longer detectable. This study highlights relevant analytical differences between SARS-CoV-2 qRT-PCR assays, leading to divergent decisions about the mandatory isolation of infected individuals. Secondly, we propose a strategy to harmonize qRT-PCR assays to achieve better comparability. Our findings are of particular interest for laboratories utilizing different assays.

## 1. Introduction

In March 2020, the World Health Organization (WHO) declared the coronavirus disease 19 (COVID19) caused by the severe acute respiratory syndrome corona virus 2 (SARS-CoV-2), a pandemic. To detect the presence of SARS-CoV-2 effectively, rapidly and with high sensitivity, the quantitative reverse transcriptase-polymerase chain reaction (qRT-PCR) is still the gold standard diagnostic tool [1,2]. In addition to detecting the virus, qRT-PCR also allows for estimation of virus quantity in the sample based on cycle threshold (Ct) values. Ct values have been shown to correlate with the initial amount of SARS-CoV-2 RNA in the PCR reaction [3,4]; and therefore, laboratories typically report Ct values in addition to the qualitative positive or negative results. Ct values can serve as an estimate of viral load in patient samples and consequently, provide information about the patient’s potential infectiousness [5,6]. Decisions to isolate or discharge patients with COVID19 are based on these values, making them very important for public infection control measures [7].

However, analytical aspects such as differences between qRT-PCR methods and assays are often not taken into account when interpreting Ct values [8]. The present study was designed to identify such analytical differences between five commonly used SARS-CoV-2 diagnostic qRT-PCR methods. Here, qRT-PCRs for SARS-CoV-2 were performed on oropharyngeal swabs of 85 COVID19 patients treated at the University Medical Center Göttingen (UMG) emergency department, using two semi-automated assays: genesig^®^ and RIDA^®^GENE as well as three fully automated methods, cobas^®^ 6800, Alinity m and GeneXpert. By comparing these five commonly available methods, our aim was to provide new analytical insights as well as contribute to a more individualized interpretation of qRT-PCR results for SARS-CoV-2 for each assay investigated. In addition, we present a strategy for harmonizing the different methods; thus, allowing better comparability between them. 

## 2. Materials and Methods

The use of left-over material for this study was covered by broad informed consent as approved by the ethics committee of the UMG (no.: 9/3/22).

### 2.1. Samples

In total, 85 oropharyngeal swabs from the UMG emergency department were analyzed for this study. Of these, 51 positive and 34 negative samples were identified using Alinity m (routinely used first at the UMG for SARS-CoV-2 RNA samples) and GeneXpert. Inactivating cobas^®^ PCR medium (Roche, Switzerland) was used as a transport medium. After initial analysis, the samples were stored at 4 °C until they were aliquoted to produce five sets of 85 samples with a volume of 1 mL each. The samples were frozen at −80 °C and thawed to room temperature prior to measurement with the respective systems. When two target assays produced discordant results, the samples were assigned as positive according to measurements using our reference assay on the Alinity m.

### 2.2. Sample Pool for Imprecision Measurements

To determine the potential imprecision of the five assays, we performed triplicate measurements on five consecutive days, which required a total of 75 identical samples. Thus, a pool of patient samples from the UMG emergency department was generated (Ct = 15.42 on Alinity m) using virus inactivating cobas^®^ PCR medium (Roche, Switzerland) as a transport medium. The pool was aliquoted at a volume of 1 mL and frozen at −80 °C until the day of measurement.

### 2.3. Samples for Standardization

Comparable to the WHO-standard RNA, a sample of a defined viral load with 10^7^ copies/mL was purchased from the external quality assessment institution INSTAND e.V. (Duesseldorf, Germany) [2]. Moreover, 10-fold serial dilutions were prepared to generate a standard curve with the respective equation. These were used for viral load concentration calculations based on the respective Ct values.

The impact of repeated freeze and thaw cycles on the stability of the diluted standard was tested performing three freeze and thaw cycles with 12 h of freezing at −80 °C each.

### 2.4. Viral RNA Extraction

Viral RNA for samples analyzed with genesig^®^ and RIDA^®^GENE kits was isolated using the QIAsymphony DSP Virus/Pathogen kit on the QiaSymphony (QiaGen, Hilden, Germany) according to the manufacturer’s instructions. A total sample volume of 200 µL was used.

### 2.5. Semi-Automated Systems

#### 2.5.1. Genesig^®^ (Primerdesign Ltd., Chandler’s Ford, UK)

The genesig^®^ Real-Time PCR assay provides reagents for detection of SARS-CoV-2 viral RNA extracted from human nasal or oropharyngeal swabs and sputum. The analysis was conducted as indicated by the manufacturer’s instructions. Briefly, the kit amplifies a SARS-CoV-2-specific target in the ORF1a/b region of the viral genome and uses 8 µL of the extracted RNA for analysis. PCR and readout were performed on the LightCycler 480 II (Roche).

#### 2.5.2. RIDA^®^GENE (R-Biopharm AG, Darmstadt, Germany)

The RIDA^®^GENE SARS-CoV-2 real-time RT-PCR assay is a reagent kit that provides reagents to detect SARS-CoV-2 from suspected human nasal or oropharyngeal swabs. Samples analyzed with the RIDA^®^GENE SARS-CoV-2 real-time RT-PCR kit were processed according to the manufacturer’s instructions. The kit uses a single target strategy and requires 5 µL of the eluted RNA sample for the analysis. According to the manufacturer, a SARS-CoV-2-specific region in the E-gene is amplified during the process. PCR and readout were performed on the LightCycler 480 II (Roche). 

### 2.6. Fully Automated Systems

#### 2.6.1. Alinity m (Abbott Molecular Inc., Illinois, IL, USA) 

The Alinity m SARS-CoV-2 assay is a real-time reverse-transcriptase polymerase chain reaction (rRT-PCR) test performed on the fully automated Alinity m system. It qualitatively detects SARS-CoV-2 RNA from nasal or oropharyngeal swabs and bronchoalveolar lavage (BAL). From a total sample volume of 500 µL, the Alinity m system automatically performs sample preparation, RT-PCR assembly, amplification, detection, results calculation and reporting. The Alinity m SARS-CoV-2 assay is a dual target assay. It uses the RNA-dependent RNA polymerase (RdRp) and nucleocapsid (N) genes to detect the presence of the SARS-CoV-2 RNA in the suspected sample. In addition, a sequence unrelated to SARS-CoV-2—from the pumpkin plant Cucurbita pepo—serves as an internal process control. 

#### 2.6.2. Cobas^®^ 6800 (Roche Molecular Systems Inc., New Jersey, NJ, USA)

The cobas^®^ SARS-CoV-2 assay was used as a qualitative test on the fully automated cobas^®^ 6800 system. It is a real-time RT-PCR test intended to detect SARS-CoV-2 RNA in nasal or oropharyngeal swab specimens collected in the Copan universal transport medium system (UTM-RT), BD™ universal viral transport system (UVT), cobas^®^ PCR media or 0.9% physiological saline. In total, 400 µL sample volume is used for fully automated sample preparation, RT-PCR and detection with the cobas^®^ 6800 system. The cobas^®^ SARS-CoV-2 assay selectively amplifies a SARS-CoV-2-specific target in the ORF1a/b viral genome region and a target in the E-gene, which is specific for the pan-Sarbecovirus family. An internal control, non-homologous to the coronavirus genome, serves as a process control.

#### 2.6.3. GeneXpert (Cepheid, California, CA, USA)

The Xpert Xpress SARS-CoV-2 assay uses real-time RT-PCR technology for the qualitative detection of SARS-CoV-2 RNA in oropharyngeal swab specimens. The single-use cartridge-based Xpert Xpress SARS-CoV-2 assay is fully automated in the GeneXpert system. It performs sample preparation, amplification and detection of the target sequences from a total of 300 µL sample volume. The dual target test uses sequences of the SARS-CoV-2-specific N2-gene and non-SARS-CoV-2-specific region of the E-gene to detect SARS-CoV-2 RNA.

### 2.7. Statistical Analysis

Statistical analysis, calculations and visualizations were conducted using the ggplot2 package [9] in RStudio [10] with R v.4.0.3 and Microsoft Excel (Microsoft, Redmond, WA, USA). For comparison of the data produced by the different methods, the Passing–Bablok regression analysis was applied. To show statistical significance the mixed-effects model was used.

Cohens kappa coefficient was calculated to analyze the agreement of the four assays to our reference (Alinity m) [11]. Additionally, sensitivity, specificity, positive and negative predictive values were determined.

## 3. Results

### 3.1. Comparability between Different Methods

At the interdisciplinary UMG laboratory, urgent samples to be tested for SARS-CoV-2 RNA are typically run on the Alinity m system; thus, Alinity m was selected as the reference method throughout this study. Qualitative comparison of the different methods revealed nearly perfect agreement between the fully automated methods; i.e., the Alinity m and the cobas^®^ 6800 (98.04%, κ = 0.98) as well as the GeneXpert (100.00%, κ = 1.00) (Table 1, upper panel). Moderate and fair agreement was observed between Alinity m and the semi-automated assays RIDA^®^GENE (90.20%, κ = 0.88) and genesig^®^ (70.59%, κ = 0.66), respectively (Table 1, lower panel). Especially the semi-automated assays showed a lower sensitivity, which was reflected by an increased number of false negative results (15 for genesig ^®^ and 5 for RIDA^®^GENE). Detailed information on sensitivity, specificity, positive and negative predictive values, based on results from our reference method (Alinity m), can be found in Appendix A.

Quantitative comparison of the Ct values revealed high variations between all the methods analyzed in this study (Figure 1A). All of these differences reached statistical significance. We found median Ct values ranging from 22.76 (Alinity m) to 31.50 (genesig^®^). This difference in the level of SARS-CoV-2 RNA would be estimated to be more than 400 times higher on the Alinity m than the genesig^®^ assay, which may lead to different clinical interpretations as well as to different public health (isolation) consequences. A list of all the Ct values is provided in Appendix A. Passing–Bablok regression analysis revealed excellent correlation coefficients (r = 0.94 to r = 0.988) with acceptable slopes ranging from 0.804 to 1.046 for cobas^®^ 6800 and GeneXpert, respectively, when compared with our reference method on the Alinity m (Figure 1B–E). However, the y-axis intercepts—indicating systematic differences—were relatively large, ranging from 3.18 for GeneXpert to 13.26 for genesig^®^.

We additionally assessed the imprecision of each method according to the Clinical and Laboratory Standards Institute (CLSI) EP05-A3 protocol. A pool of patient samples was generated and split into 75 identical samples. These were measured as triplets on five consecutive days using each method. Total imprecision ranged from as little as 0.13 Ct values on the cobas^®^ 6800^®^ to 1.68 Ct values for the genesig^®^ assay (Table 2). Within-run imprecision was generally much lower and showed a similar pattern between the different methods; i.e., highest for the genesig^®^ assay (Table 2). In general, all the methods were found to be fairly precise, with the fully automated test systems performing better than the semi-automated assays.

### 3.2. Harmonization of Different SARS-CoV-2 qRT-PCR Assays

In the next step, we aimed to overcome the large variations between the different methods, shown in Figure 1, to allow comparability of SARS-CoV-2 RNA quantifications. Thus, we diluted a sample of defined viral load to generate standard curves for each method [2]. The sample had a known concentration of 10^7^ copies/mL and was diluted stepwise (1:10) to generate a total of six samples with decreasing concentrations. Measurements were then plotted against the respective Ct values to create five individual standard curves with respective equations to calculate individual virus concentrations (Figure 2). According to our measurements, a concentration of 10^6^ copies/mL corresponds to Ct values of 21.61, 26.41, 27.04, 31.04 and 29.75 using the Alinity m, cobas^®^ 6800, GeneXpert, genesig^®^ and RIDA^®^GENE assays, respectively (Appendix A). Notably, concentrations of 10^3^ and 10^2^ copies/mL could not be detected using the semi-automated methods. Hence, the total numbers for comparative analysis were reduced for these assays. Corresponding equations to calculate assay-specific viral load concentrations were: Alinity m: y = 9 × 10^11^ × e^−0.632x^, cobas^®^ 6800: y = 1 × 10^16^ × e^−0.875^, GeneXpert: y = 8 × 10^13^ × e^−0,673x^, genesig^®^: y = 2 × 10^15^ × e^−0.69x^, RIDA^®^GENE: y = 9 × 10^14^ × e^−0,693x^ (Figure 2). Multiple freeze/thaw cycles had no relevant impact on the generation of standard curves, as shown exemplarily for our reference method on the Alinity m (Appendix A).

The equations were then used as a basis to quantify individual viral loads for each method according to the respective Ct values. This resulted in an obvious narrowing of fold changes of all methods compared to our reference method on the Alinity m (Figure 3). Thus, the individual adjustment resulted in a much better comparability between the different methods (median log (Ct) of all methods = 5.14 vs. median log (viral load) of all methods = 0.35) (Appendix A).

Following harmonization using individual standard curves to calculate the viral load, all significant differences between the methods, as observed when comparing the Ct values as previously demonstrated in Figure 1, disappeared. (Figure 4). Median concentrations ranged from 5.10 × 10^5^ (Alinity m) to 1.70 × 10^6^ (cobas^®^ 6800) copies/mL.

For the three automated assays, our proposed strategy to harmonize different SARS-CoV-2 qRT-PCRs by quantifying methods-specific viral concentrations would have an effect on decisions for isolating SARS-CoV-2 infected individuals. In more detail, our approach—as compared to pure consideration of the raw Ct values—would result in a reduction of isolations, when the often-used cutoffs (Ct < 30 and viral load concentration > 10^6^ copies/mL) are applied [3,4,5,6] (Appendix A).

## 4. Discussion

Ct values serve as the main parameter for semi-quantitative assessment of viral load and have also been used as a surrogate marker for the amount of infectious viral particles [3,4,5,6]. However, Ct values are affected by several variables, including pre-analytical factors, such as sampling time, site and method [7,8,12]; the specimen type and patient age; [13] as well as transportation medium and temperature. In addition, assay-specific analytical factors, such as the target sequence of the PCR reaction, the presence of PCR inhibitors as well as primer efficiency may all have significant influences on the Ct value [8]. In the ongoing SARS-CoV-2 pandemic, different qRT-PCR methods and assays are available and may even be used within the same laboratory. This highlights the need to study and better understand differences between different qRT-PCR assays. 

Previous studies on molecular diagnosis of SARS-CoV-2 demonstrated very high concordance of three fully automated systems; i.e., cobas^®^ 6800, Alinity m and Hologic Panther Fusion [14,15]. This is in agreement with our qualitative results showing very good comparability between cobas^®^ 6800, Alinity m and GeneXpert. In contrast, comparability between the semi-automated assays genesig^®^ and RIDA^®^GENE versus the Alinity m was rather poor. Despite the good agreement between the automated systems on qualitative results, we found large variations in the respective Ct values. These variations were significant between all the methods, even the fully automated ones, included in this study. Without any standardization, differences of two to four cycles among the fully automated methods were found. This, in turn, corresponds to an approximately 10-fold difference in the respective viral load concentrations. Ultimately, this may lead to different clinical interpretations, especially when deciding on isolation for SARS-CoV-2 infected individuals. There is thus an urgent need to standardize quantitative SARS-CoV-2 RNA measurements, when Ct values or viral concentrations are reported. 

In this study, we demonstrate an approach for overcoming this problem and harmonizing different qRT-PCR methods for assessing SARS-CoV-2 viral load. Our approach comprised of measuring a sample of a defined SARS-CoV-2 concentration in a serial dilution using all five methods, followed by the generation of individual standard curves and respective equations [2]. These equations were then used as a basis to calculate individual viral loads. Our results show that this approach harmonizes the different methods and improves their comparability. This strategy may also be useful for other laboratories using different qRT-PCR methods and assays. In fact, based on the data from this study, we now regularly provide the method-specific viral load for each SARS-CoV-2 report at the University Medical Center Göttingen.

Standard curve generation showed that low viral loads were not detected by semi-automated approaches. Our data indicate that the lowest Ct values were measured by the Alinity m system, which showed the highest sensitivity among all the methods. In part, this might be due to the fact that the Alinity m uses two target genes and detects both of them in the same fluorescence channel to produce only one Ct value. This will ultimately lead to a stronger fluorescence signal and result in increased sensitivity compared to the other methods.

In addition, viral variant type may also impact variations in SARS-CoV-2 qRT-PCR diagnostics. Here, we focused on the inter-device variability rather than differences caused by viral mutations. In times of continuous viral adaptation, we provide a simple and straightforward method to easily harmonize different assays. In order to overcome lot-specific and run-to-run variations, defined viral load standards should be included in every qRT-PCR run, followed by the generation of individual standard curves. These will then provide the basis for respective viral load concentration calculations.

The variability of Ct values in SARS-CoV-2 diagnostics has also been noted in external quality assessment studies [4,16,17] which, as our findings also show, highlight the limitations of Ct value interpretation from SARS-CoV-2 qRT-PCR. Using standardized samples with a defined viral load, we could demonstrate that all of the automated systems produced results far below the suggested Ct cutoff value of 30 that predicts the likelihood of infectivity of the patient [18,19,20]. This indicates a higher sensitivity of these assays that not only leads to isolation of the patient (with lower viral loads), but also extends the length of quarantine due to prolonged Ct value PCR-positivity below the cutoff. Taken together, the harmonization of qRT-PCR results by calculation of viral loads reduced variations seen in Ct values. Interpretations of the results were thus more comparable among measuring systems and diagnostic laboratories, especially in longitudinal analysis of individual patients.

## 5. Conclusions

In conclusion, we demonstrate significant variations in the Ct values of different SARS-CoV-2 qRT-PCR assays, especially between fully and semi-automated ones. In an effort to harmonize and reduce these variations, we performed a standardization measurement on defined viral load concentration samples and suggested formulas for the approximation of viral loads from the Ct values for Alinity m, cobas^®^ 6800, GeneXpert, genesig^®^ and RIDA^®^GENE assays. Our results suggest that standardized viral load concentrations rather than Ct values should be reported in order to harmonize SARS-CoV-2 qRT-PCRs reports and interpretations.

## Figures and Tables

**Figure 1 viruses-14-02239-f001:**
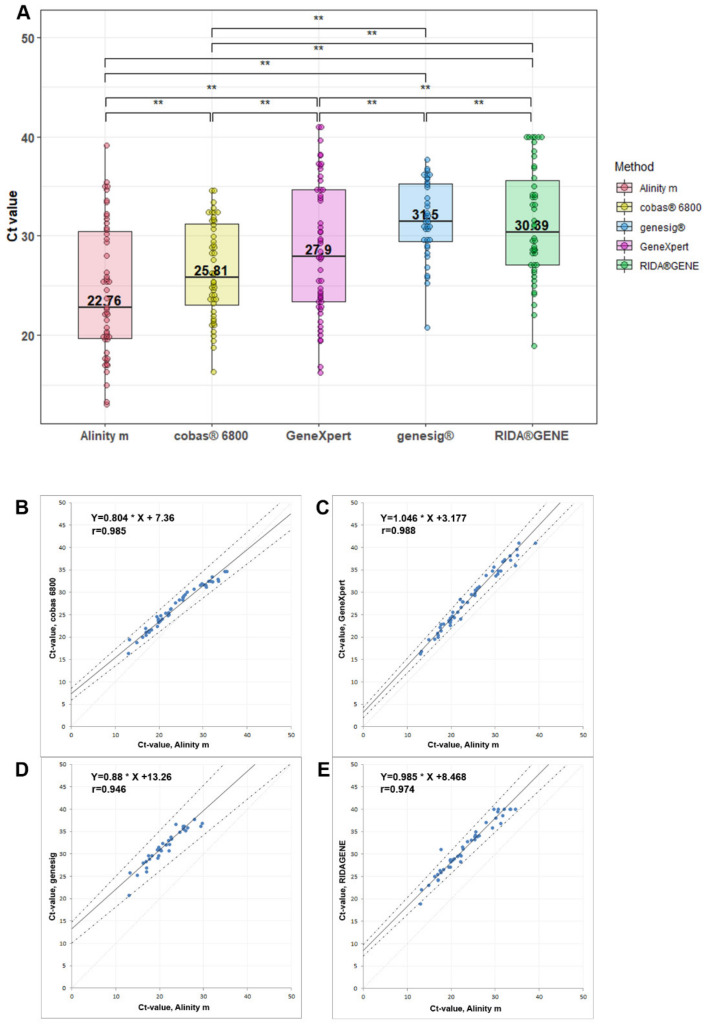
Overview of the Ct value ranges (**A**) and Passing-Bablok regression analyses (**B**–**E**). (**A**) Median Ct values and 95% confidence intervals are shown for the different methods. Significance (** = *p* < 0.01) between different methods using mixed-effects analyses followed by Tukey´s multiple comparison test. (**B**–**E**) Passing–Bablok regression analyses vs. our reference method on the Alinity m. Linear equations and correlation coefficients (r) are reported.

**Figure 2 viruses-14-02239-f002:**
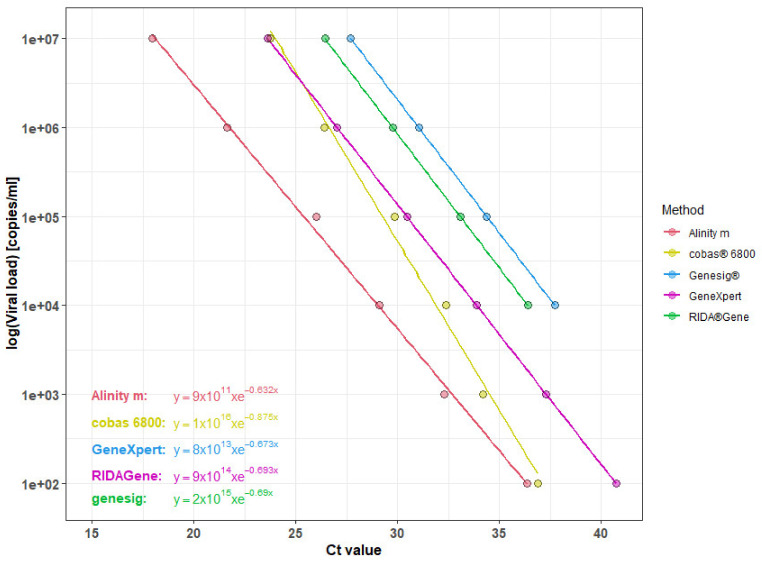
Individual standard curves for the different methods. Serial dilutions of a known SARS-CoV-2 concentration of 10^7^ copies/mL were performed using each of the five methods. Ct values were then plotted against the respective viral loads and individual standard curves with according equations for viral load quantification were generated.

**Figure 3 viruses-14-02239-f003:**
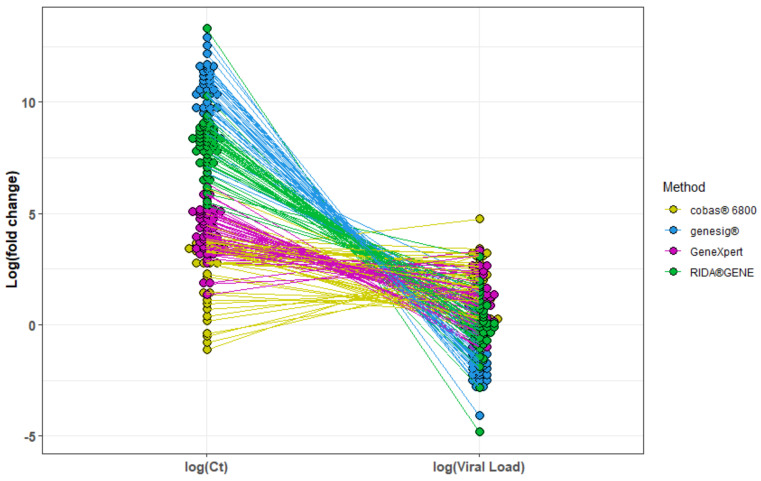
Variations before and after our proposed harmonization. Log fold changes before (according to Ct values, log(Ct)) and following individual viral load calculations (log(Viral Load)) between the four methods vs. our reference method (Alinity m) are shown for each sample.

**Figure 4 viruses-14-02239-f004:**
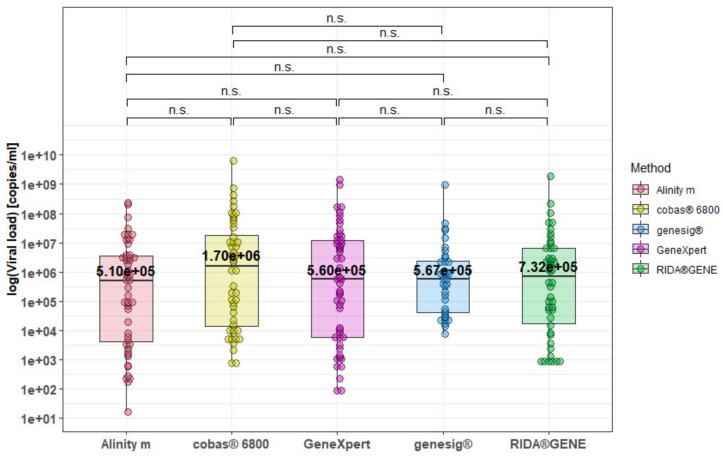
Range of viral loads following individual adjustment for the different methods. n.s. = not significant between different methods using mixed-effects analyses followed by Tukey´s multiple comparison test.

**Table 1 viruses-14-02239-t001:** Overall agreement (OA) between the different methods of assessing SARS-CoV-2 RNA. Qualitative positive or negative results for SARS-CoV-2 were compared between our reference method (Alinity m) and four other methods, two of which were fully automated (upper panel) and two semi-automated (lower panel). Overall agreement and the Cohen´s kappa coefficients are reported.

		Cobas^®^ 6800				GeneXpert			
		Positive	Negative	Total	OA	Positive	Negative	Total	OA
Alinity m	Positive	50	1	51	98.04%	51	0	51	100.00%
	Negative	0	34	34	κ = 0.98	0	34	34	κ = 1.00
	Total	50	35	85		51	34	85	
		**genesig^®^**				**RIDA^®^GENE**			
		**Positive**	**Negative**	**Total**	**OA**	**Positive**	**Negative**	**Total**	**OA**
Alinity m	Positive	36	15	51	70.59%	46	5	51	90.20%
	Negative	0	34	34	κ = 0.66	0	34	34	κ = 0.88
	Total	36	49	85		46	39	85	

**Table 2 viruses-14-02239-t002:** Summary of imprecision measurements among the different methods.

		Alinity m	Cobas^®^ 6800	GeneXpert	Genesig^®^	RIDA^®^GENE
Mean Ct		15.67	18.69	20.03	25.47	23.99
Imprecision:	Within Run	0.30	0.08	0.65	0.21	0.14
	Total	0.35 (%CV = 2.2)	0.13 (%CV = 0.7)	0.64 (%CV = 3.2)	1.68 (%CV = 6.6)	1.40 (%CV = 5.8)

## Data Availability

Not applicable.

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
