# Peer review of "Comparison and Harmonization of Different Semi-Automated and Automated qRT-PCR Assays in the Assessment of SARS-CoV-2"

_viruses, 2022, doi:10.3390/v14102239_

Round 1
Reviewer 1 Report
The manuscript by Dierks et al presented a comparison of performances of the current available qRT-PCR assays for detection of SARS-CoV-2. While the study meets a critical need in SARS-CoV-2 diagnosis, there are several major concerns:
1. How it was determined to use 55 clinical specimens for this study?
2. Inclusion of 55 clinical specimens may be insufficient, as the author had to make sample pools to meet the demand of a larger number of samples. Moreover, results from pooled samples may not reflect the true performance of the evaluated assays, because the number of enrolled samples were insufficient.
3. The existence of SARS-CoV-2 in clinical specimens should be confirmed by a wide accepted assay, for example, gene sequencing, rather than by an assay evaluated in the present study.
4. Could the authors provide more information on the “Assay-specific viral load concentration calculations (calibrations?)”? And is it affected by the sample size or other factor?
5. It is inappropriate to apply the judgement “When 70 two-target assays produced discordant results, the samples were assigned as positive.” in the Samples paragraph of the Materials and Methods section Indeed, a repeat test is recommended to confirm the contradictory results.
6. A wide accepted assay (for example sanger sequencing or NGS sequencing) is recommended to apply for comparison of the evaluated qRT-PCR assays. And more parameters are needed for comprehensive comparison of the assays beside the kappa index, for example, accuracy sensitivity, specificity, positive predictive value, negative predictive value.
Reviewer 2 Report
In the current manuscript, Dierks et al. investigated the difference between semi-automated and automated qRT-PCR assays to detect SARS-CoV-2 infection in patients. Furthermore, they proposed a harmonization method to allow for cross-assay comparative assessment of viral load by using a defined virus sample and making a standard curve. Overall, the manuscript is well-written, and the experimental design is logical. The data presented by the authors to support their claims are fairly convincing, however there are still some inconsistencies and omissions in the methods that need to be properly addressed. Below are my critical comments for the current study:
1. There are 51 positive SARS-CoV-2 samples and only 4 negative samples. The authors should include more negative samples to determine the false positive rate among the five assays tested. 4 negative samples are not sufficient.
2. In Fig. 2, why is the y-axis so arbitrary (i.e. 3.27, 1.63, etc.)? If this the standard curve dilution, please plot the exact number of viral load (copies/ml) for each point.
3. The authors obtained a sample of defined viral load and performed serial dilution (1:10) to generate a standard curve. It would be interesting to test if freeze/thaw cycle will affect the Ct values of the standard curve or not. Since this paper proposed a method to harmonize the Ct values across different assays, this experiment would be helpful for other labs if they decide to adopt this method proposed by the authors. I suggest the authors perform 5 freeze/thaw cycles to see if this will degrade the standards in any significant way.
4. Additionally, did the authors analyze the standards at the same time as the patient samples or did they just run the standards separately to generate an equation and then use the equation retroactively to calculate the viral load in the patient samples with the Ct values from previous run? If this is the case, it may lead to incorrect viral load estimation since the Ct values can vary among different qPCR experiment. Ideally, the standards should be included in the same run as the unknown samples.
5. In lines 81-82, the authors wrote: “…from the external quality assessment institution INSTAND e.V. (Duesseldorf, Germany) (9)”. They cited reference number 9, but upon checking, reference #9 has nothing to do with this standard. Please double-check.
6. In line 205, “…concentration of 107 copies/mL…” the 7 should be superscript.
7. In line 221, “5.10*105” and “1.70*106” both need superscript.
8. The authors should move the last paragraph of the Discussion section to the Conclusions section since that paragraph started with “In conclusion” and the Conclusions section is empty.
Round 2
Reviewer 2 Report
The authors have adequately addressed all of my concerns. The paper can be accepted in its current form.